# Designing Multi-Stage 2 A/O-MBR Processes for a Higher Removal Rate of Pollution in Wastewater

**DOI:** 10.3390/membranes12040377

**Published:** 2022-03-30

**Authors:** Zhengzhong Zhou, Bin Zhang, Qian Wang, Xiaoshan Meng, Qigang Wu, Tao Zheng, Taoli Huhe

**Affiliations:** 1National-Local Joint Engineering Research Center of Biomass Refining and High-Quality Utilization, Institute of Urban and Rural Mining, Changzhou University, Changzhou 213164, China; 13151232973@163.com (B.Z.); chandler@cczu.edu.cn (Q.W.); xshmeng@cczu.edu.cn (X.M.); wqg1990@cczu.edu.cn (Q.W.); zhengtao@ms.giec.ac.cn (T.Z.); hhtaoli@cczu.edu.cn (T.H.); 2Guangzhou Institute of Energy Conversion, Chinese Academy of Sciences, No. 2 Neng Yuan Road, Tianhe District, Guangzhou 510640, China

**Keywords:** membrane bioreactor, wastewater treatment, microbial community, membrane fouling, multi-stage A/O

## Abstract

Multi-stage A/O-MBR processes were designed to improve wastewater treatment efficiency; three different designs were carried out and compared in this study. The 2(A/O)-MBR process, i.e., with two sets of anoxic/oxic tanks in series, showed better effluent quality than A/O-MBR and 3(A/O)-MBR processes. The removal rates of COD, NH${}_{4}^{+}$-N, TP and TN were 95.29%, 89.47%, 83.55% and 78.58%, respectively, complying satisfactorily with China’s urban sewage treatment plant pollutant discharge standards. In terms of membrane fouling, the 3(A/O)-MBR process demonstrated the lowest fouling propensity. The microbial community structure in each bioreaction tank was analyzed, the results from which matched with the process efficiency and fouling behavior.

## 1. Introduction

The severe shortage of water resources on a global scale is one of the greatest challenges mankind is facing. As the economy develops, people’s demand for water resources in daily life is also increasing [1,2], such that a larger amount of sewage is inevitable. Sewage has a relatively complex composition due to the wide range of sources. If not handled carefully and discharged without treatment, the contaminants, including the relatively high nitrogen and phosphorous content, can cause groundwater pollution, the eutrophication of lakes, rivers and other water bodies, and the proliferation of planktonic algae [3], having a serious impact on water resources and the ecological environment. Technologies for the proper treatment and recycling of sewage are thus needed [4]. Membrane bioreactor (MBR) is a sewage treatment process that combines biochemical treatment process with membrane separation technology. It demonstrates various advantages, such as excellent effluent water quality, efficient solid–liquid separation capacity, and low sludge output [5,6,7,8,9], therefore it has been widely applied in sewage treatment.

MBR is often coupled with anaerobic and aerobic bioprocesses to effectively remove nitrogen and phosphorus [10,11]. Studies have found that the A/O-MBR and A^2^O-MBR process [12,13] can effectively improve the nitrogen and phosphorus removal efficiency. Sun et al. designed an integrated A/O-MBR [14] to treat highly saline urban sewage. The results show that the removal efficiency of the system for ammonia nitrogen, total nitrogen and TP is 95%, 50–70% and 60–80%, respectively. The reactor has a poor impact load capacity for nitrogen. When there is a high load of nitrogen impact, the system’s removal rate of ammonia nitrogen and total nitrogen will decrease severely. Meanwhile, the reactor relies on biological phosphorus removal, and when the reactor is running for a long time, the biological phosphorus removal effect is not good [15]. Poly-Aluminum chloride (PAC) can effectively improve the phosphorus removal effect [16], but the addition of PAC will reduce the denitrification capacity of the A/O-MBR system [17]. Zhang et al. studied the removal effect of a reverse A^2^O-MBR system on COD, etc. [18]. The results show that the system has a high removal rate of COD and ammonia nitrogen, but the system has strict requirements for aeration, as, otherwise, it causes the TMP to rise rapidly and membrane fouling. Falahti-Marvast et al. used a pilot scale A^2^O-MBR to treat municipal wastewater [19]. The results show that the system has a high removal rate of COD and TN, but a low removal rate of TP.

The above studies have effectively improved the nitrogen and phosphorus removal efficiency of the MBR system in several cases, but they are not able to cope competently with the fluctuating wastewater composition. Zheng et al. found that the multi-stage A/O process can increase the diversity of microorganisms in the system to improve the impact resistance of the system effectively [20]. Wang et al. found that the multi-stage A/O process can improve the nitrogen removal ability under low carbon sources [21]. Considering the advantages and disadvantages of the above works, this research integrates the Multi-stage A/O process and the MBR process to design an overflow multi-stage A/O-MBR processes. Alternating A/O operations with various stages are combined with ultrafiltration membranes for enhanced stability and a higher removal of pollutants, such as nitrogen and phosphorus.

In this study, the efficiencies in treating sewage by various combinations are compared first, followed by the analysis on membrane fouling behaviors, and finally, the microbial community structure in each reaction tank under different process conditions. As such, this work aims not only to provide the design of an innovative and efficient wastewater treatment system, but also an explanation regarding the underlying mechanism in terms of microbial community changes.

## 2. Materials and Methods

### 2.1. Materials and Reagents

Glucose, magnesium sulfate, zinc sulfate and sodium chloride were purchased from Chinasun Specialty Products Co., Ltd., Suzhou, China. Urea and potassium dihydrogen phosphate were purchased from Shanghai Lingfeng Chemical Reagent Co., Ltd., Shanghai, China. Manganese sulfate, iron sulfate, calcium chloride, and peptone were purchased from Shanghai Maclean Biochemical Technology Co., Ltd., Shanghai, China. Cobalt chloride and beef powder were purchased from Shanghai Aladdin Biochemical Technology Co., Ltd., Shanghai, China.

Hollow fiber membrane modules made of polyvinylidene fluoride (PVDF) were purchased from Coriolis (Beijing) Membrane Technology Co., Ltd., Beijing, China. The nominal pore size and the effective area of the membrane is 0.1 μm and 0.726 m^2^, respectively. The activated sludge used in the experiment came from a sewage treatment plant in Changzhou, and the simulated wastewater uses glucose (C_6_H_12_O_6_) as the carbon source, urea (CO(NH_2_)_2_) as the nitrogen source, and potassium dihydrogen phosphate (KH_2_PO_4_) is used to provide phosphorus. Its actual composition is shown in Table 1. After measurement, the actual values of COD, NH${}_{4}^{+}$-N, TN and TP of artificial wastewater are about 400 mg/L, 40 mg/L, 70 mg/L and 6 mg/L, and the ratio of COD:TN:TP of simulated sewage is 400:70:6.

### 2.2. Experimental Setup

The experimental device is shown in Figure 1. The reactor was made of acrylic material, including the inlet tank (16 × 32 × 50 cm), anoxic tank (16 × 16 × 50 cm), and aerobic tank (16 × 32 × 50 cm) and membrane tank (32 × 32 × 50 cm); the volume ratio of the inlet tank: anoxic tank: aerobic tank: membrane tank was 2:1:2:4.

### 2.3. Test Methods and Operating Parameters

The simulated sewage was injected into the inlet tank through a peristaltic pump (BT100J-1A, Huiyu Weiye (Beijing) Fluid Equipment Co., Ltd., Beijing, China). The sewage flowed from the inlet tank to the aerobic tank, to the membrane tank in the form of overflow, and finally was returned to the inlet tank by the peristaltic pump. Figure 1 shows, from top to bottom, A/0-MBR, 2(A/O)-MBR and 3(A/O)-MBR. The sewage was pumped out by the diaphragm pump (Ningbo Leicheng Pump Co., Ltd., Ningbo, China) through the membrane module and discharged into the collecting tank. The effective volumes of the three processes were 112.5 L, 150 L and 187.5 L, respectively.

The experiment was carried out in the following mode: operation for 10 min followed by pausing for 2 min; backwashing with water for 30 s after 15 cycles (180 min). The permeability of hollow fiber membranes decreased with the proceeding of filtration, so the power of the diaphragm pump was adjusted to increase the trans-membrane pressure; the water flux was thus kept roughly constant at 20 L/(m^2^ h). Each process was conducted continuously for 20 d at room temperature (about 25 °C), then the hollow fiber membrane module was soaked in a 0.1 mol/L NaClO solution for 2 h. The sludge retention time (SRT) was 20 d, and the blower continuously aerated the aerobic tank and membrane tank at a flow rate of 0.65 L/d to keep the dissolved oxygen (DO) of the aerobic tank and membrane tank at about 2.5 mg/L. The DO of the anaerobic tank was kept below 0.3 mg/L, and the DO of the inlet tank was in the range of 1–2 mg/L. The recirculation ratio of the sludge mixture was 200%, the activated sludge concentration (MLSS) in the aerobic zone was maintained at 5.0 ± 0.5 g/L, and the MLSS in the anoxic zone was maintained at 8.0 ± 0.5 g/L throughout the experiment.

### 2.4. Analytical Method

#### 2.4.1. Sewage Analysis

The COD, NH${}_{4}^{+}$-N, TP and TN of the wastewater in the inlet, anoxic, aerobic and membrane tank were analyzed by standard methods with the aid of an ultraviolet-visible intelligent multi-parameter water quality analyzer (Lianhua Environmental Protection Technology Co., Ltd., Lanzhou, China) every day. Before characterization, the sample was centrifuged by Centrifuge 5810R (Eppendorf China Ltd., Shanghai, China) and the supernatant was extracted to test. MLSS, pH and DO were measured by the gravimetric method, acidity meter FE20-K (METTLER TOLEDO, Shanghai, China) and dissolved oxygen meter JPB-607A (INESA Scientific Instrument Co., Ltd., Shanghai, China), respectively.

#### 2.4.2. Membrane Fouling Analysis

The morphology the membranes before and after filtration was analyzed by scanning electron microscope (SEM, Zeiss supra-55, Carl Zeiss AG, Oberkochen, Germany). Before characterization, the sample with length of 10 cm was cut from the membrane module and soaked in deionized water for 2 days. Then, the residual water on the membranes was removed with the wiping paper, and the membranes were frozen in a low-temperature refrigerator at −20 °C for 40 min. After that, the samples were dried in a freeze dryer. Prior to the observation, the samples were sputtered with nano-gold.

During the experimental process, the flux of the membranes was tested as follows: the permeate water was collected for 30 min at a fixed time every day, and the flux was calculated by the Equation (1):(1)$$J=\frac{V}{t\times A}$$
where *J* is the membrane flux (L/(m^2^ h)), *V* is the permeate volume (L), *t* is the filtration time (h) and *A* is the effective area of the membrane (0.726 m^2^). Meanwhile, the transmembrane pressure difference (TMP) is recorded every 6 min to obtain five values TMP_1_~TMP_5_, and the average value was used as the TMP of this experiment.

#### 2.4.3. Analysis of Microbial Community Structure

Sludge samples were taken from the inlet tank, anoxic tank, aerobic tank and membrane tanks. An DNA extraction kit (QIAGEN Gmbh, Hilden, Germany) was used to extract the DNA samples from the sludge. The front primer: 343F-5′-TACGGRAGGCAGCAG -3′ and back-end primer: 798R-5′-AGGGTATCTAATCCT-3′ were applied using an Illumina Miseq sequencer to complete DNA sequencing. The phylum, class, and genus were comprehensively compared among the top 15 species in the ranking of abundance, and the differences in the microbial community structure in the sludge in the reaction tanks of different processes were analyzed. Alpha diversity analysis, including diversity index dilution curve, diversity index boxplot analysis and Rank Abundance analysis, was applied to reflect the diversity of microorganisms in the research sample comprehensively.

## 3. Results and Discussion

### 3.1. Effective Removal of Pollutants

Figure 2 shows the overall removal rate of COD, NH${}_{4}^{+}$-N, TP, and TN by the three processes, and Figure 3 shows the variation of pollutant concentrations in individual tanks for each process. In general, all three processes demonstrate the high removal effect of the four pollution indicators, suggesting the feasibility of the combination of AO and membranes processes. The comparation of the specific removal performance of the three processes are discussed as follows.

As shown in Figure 2a, the COD values of the permeate from the A/O-MBR, 2(A/O)-MB and 3(A/O)-MBR process are 40.87 mg/L, 20.02 mg/L and 37.78 mg/L, and the average removal rates are 90.29%, 95.29% and 90.99%, respectively, all complying with the Sewage Disposal Standard Class 1A.

As the inlet water flow rate and reflux ratio remain unchanged, the HRT of the three processes increases with more A/O stages, but the COD removal rate of the three processes is similar, which shows that HRT does not have a great impact on the COD removal effect of the current processes. Relatively speaking, the 2(A/O)-MBR process design shows the highest COD removal rate, which implies that increasing the number of A/O stages not necessarily guarantee a better COD removal. Ji et al. [22,23] shows that the microbial degradation in the sludge reaches an optimum state with a suitable HRT, and the further increase in HRT may adversely affect the effluent water quality. This is in good agreement with our result that the COD removal rate of the 3(A/O)-MBR is lower than that of the 2(A/O)-MBR process.

The biological denitrification process mainly includes two processes: the nitrification by autotrophic bacteria under aerobic conditions and the denitrification by heterotrophic bacteria under hypoxic conditions [24,25,26,27]. Figure 2b,d show the difference in the nitrogen removal efficiency of various processes. The average removal rates of NH${}_{4}^{+}$-N and TN by the 3(A/O)-MBR process are 58.44% and 39.83%, respectively, and those by A/O-MBR are 80.36% and 57.19%, respectively; both are less than the efficiencies of the 2(A/O)-MBR process, of which the average removal rates are 89.47% and 78.58%, respectively. The above results suggest again that the number of A/O stages in series is not directly proportional to treatment efficiency. Two possible reasons may be responsible for the difference. On the one hand, the 3(A/O)-MBR process has more reaction tanks in which the pollutant concentration decreases in series, and the carbon source in the latter reaction tanks may be too low for the denitrification process, resulting in a poorer total nitrogen removal. On the other hand, the lower total HRT and rapid recirculation in the A/O-MBR process may lead to DO rises up to 0.5–1 mg/L in the anoxic pool, which exceeds the DO range of strict hypoxia (0.3–0.5 mg/L) and reduces the denitrification performance. Thus, the denitrification efficiency is relatively the highest in the 2 (A/O)-MBR setup.

Different from the nitrogen removal capacity, the average removal rates of TP by the three processes were 78.91%, 83.55% and 90.04%, respectively. As shown in Figure 2c, the 3(A/O)-MBR process not only shows the lowest TP concentration in the effluent, but also the most stable values, suggesting that the more A/O stages may result in a better TP removal efficiency. This may be because of the extra aerobic tanks, where the phosphorus-accumulating bacteria function by absorbing the phosphorous from wastewater in the n(A/O)-MBR system. As the total amount of phosphorus-accumulating bacteria increases, the TP removal efficiency increases as well.

In general, the pollutant concentrations in the effluent comply well with the sewage discharge standard (Class I A Discharge Standard, GB18918-2002, China), namely, the concentrations of COD, NH${}_{4}^{+}$-N, TP and TN are lower than 50.0 mg/L, 5.0 mg/L, 0.5 mg/L and 15.0 mg/L, respectively. This demonstrates that it is feasible to use the overflow O-MAO-MBR to treat sewage. Further, by evaluating the removal rates of COD, NH${}_{4}^{+}$-N, TP and TN, the 2(A/O) -MBR process is more suitable than the other two processes.

### 3.2. Membrane Fouling

The test period of each process is 20 d. As the same membrane module is used for the three types of setups, the initial permeabilities and TMP of all three processes are normalized according to that of A/O-MBR. The corresponding results of the three processes are shown in Figure 4. The trends in TMP are similar; the gradient of the TMP curves follows a decreasing then increasing path. Similarly, the trends in permeability follow the same trend too. However, in terms of the absolute values, the final TMP of the A/O-MBR process at the end of experiment is the largest and the corresponding permeability is the lowest among the three processes, while the 3(A/O)-MBR process demonstrates the smallest TMP and the highest permeability. The result suggests that membrane fouling decreases with increasing the number of A/O stages. Studies [28,29] have shown that a high organic load aggravates the membrane fouling in a MBR process, and reducing the organic load can effectively alleviate membrane fouling. Among the three processes, the effective total working volume of the reactor increases as the number of A/O stages increases, such that the organic load of the system deceases, which could be the main reason for the slightly lower fouling propensity.

The membrane is chemically cleaned with NaClO after each test. The TMP of the three membrane modules after cleaning drops to 0.301 bar, 0.296 bar and 0.290 bar. Chemical cleaning is an effective way to restore the membrane, which is consistent with the research results of Zhang et al. [30]. However, the TMP is not fully restored to the initial value 0.278 bar, suggesting the foulant in the PVDF membrane is not completely removed. The reason may be due to the irreversible fouling that the pollutants stacked in the membrane pores or matrix reduce the effective pore sizes and partially block the water pathways [31,32]; as a result, the membrane flux decreases.

Figure 5 shows the surface of the pristine membrane (a) being fouled (b), cleaned (c), fouled again (d), cleaned again (e) and finally fouled the third time (f). A clear hole structure can be observed on the pristine membrane surface, and sever fouling (Figure 5d–f) is seen on the membrane surface after operation. Figure 5b,c demonstrate that most of the foulants are removed by chemical cleaning; however, the surface pores are still covered with some contaminants, as compared against Figure 5a. This result echoes with the transmembrane pressure difference after cleaning above, indicating that irreversible membrane fouling is present in the membrane module even after chemical cleaning.

### 3.3. Microbial Community Structure Analysis

Figure 6a is the microbial community structure column diagram of the sludge samples at the phylum level from the three processes. The samples are numbered according to Table 2. The top 15 species of microbial abundance in the sludge in all reaction tanks of the 3 processes are the same. Proteobacteria, Bacteroidetes, Firmicutes and Actinobacteria account for a total of about 90%, with individual average values of 53.06%, 22.22%, 9.64% and 6.30%, respectively. The above data indicates that the Proteobacteria and Bacteroides are the dominant flora in the reactor, consistent with the study of Tang et al. [33,34].

Figure 6b shows the top 15 species of abundance at the class level. γ-Proteobacteria, Bacteroidetes, α-Proteobacteria, δ-Proteobacteria and Bacilli are the dominant bacteria, corresponding to the dominant bacteria classes under the Proteobacteria, Bacteroides and Firmicutes. γ-Proteobacteria shows the highest abundance at the class level in this study, and its variation trend was the same as that of Proteobacteria. According to the Berger bacterial identification manual, γ-Proteobacteria can utilize NH${}_{4}^{+}$ as the nitrogen source and glucose as the carbon source, metabolize glucose to produce acid and reduce NO^−3^ to NO^−2^ [35,36]. This indicates that γ-proteobacteria may be one of the main ammonia-oxidizing bacteria (AOB). Kim et al. [37] showed that nitrifying bacteria (Nitrobacteria) were the main nitrite-oxidizing bacteria (NOB) in nitrification, while the results of Zhang et al. [38] suggested Nitrospira can oxidize NO^−2^ to NO^−3^, acting as the main NOB in the nitrification reaction. The reason may be the rapid growth of Nitrospira under low nitrite concentration [37]. In addition, the changes of Bacilli in each reaction tank of different processes are different. In the A/O-MBR and 2(A/O) -MBR processes, the abundance of Bacillus was higher than that in the 3(A/O) -MBR process. The reason is possibly due to the high resistance of Bacillus to external harmful factors in comparison to other species. In the A/O-MBR and 2(A/O)-MBR processes, the impact load is relatively large, and the adaptability of the bacillus is relatively stronger, so its proportion is relatively high. In the 3(A/O)-MBR process, the impact load is relatively low, and the proportion of other bacteria classes increases, such that the relative proportion of Bacillus species decreases.

Figure 6c shows that, at the genus level, the A/O-MBR and 2(A/O)-MBR processes have a higher abundance of Trichococcus in the inlet pool. It gradually decreases to 5.92% and 5.23% in the flow direction. However, its abundance is much lower in the 3(A/O)-MBR process, only about 0.28%. Trichococcus is a chemotrophic heterotrophic filamentous bacterium that can cause sludge bulking [39]. According to the report [40], sludge bulking is one of the causes for membrane fouling. The abundance of Trichococcus in the 3(A/O)-MBR process is lower than that of the other two processes; this may be one of the reasons for the relatively lower membrane fouling. Meanwhile, the genus trichococcus is a chemotrophic bacterium, and the gradual reduction in its abundance along the flow direction can also reflect the gradual decrease in organic matter in the reaction tanks.

Denitratisoma, Flavobacterium and Thauera are important denitrifying bacteria from the samples in this study. Among them, Thauera is an autotrophic denitrifying bacterium belonging to the phylum Proteobacteria. It is particularly valued for its ability to degrade aromatic organic compounds [41]. The average abundance of Thauera is 1.33%, 1.54% and 0.43% in the three processes. In addition, according to the data in Figure 2, the removal rates of ammonia nitrogen by the three processes were 80.36%, 89.47% and 58.44%, respectively, and the removal rates of total nitrogen were 57.19%, 78.58% and 39.83%. The nitrogen removal effect of each process is consistent with the value of Thauera abundance, indicating that Thauera has a very important role in the removal of nitrogen in the reactor. This result is also consistent with that from other researches [42,43,44,45].

Table 3 presents the statistical data of the alpha diversity index. Figure 7a is the Simpson dilution curve. It shows that when the number of sampled sequences increases, the Simpson number reaches the maximum value quickly. This result indicates that the sampling meets the sequencing depth and that the sample sequencing amount is reasonable and has covered all groups.

Figure 7b shows the relationship among the differences in microbial diversity in groups A, B, and C, where S_DA_ > S_DB_ > S_DC_. That is, as the number of A/O stage increases, the difference in microbial diversity in each reaction tank in the three processes gradually reduces. Additionally, from the middle line of the box plot, it can be seen that the diversity index gradually becomes larger with more A/O stages, that is, S_A_ < S_B_ < S_C_. The microbiological difference between group A and group B was not significant, but group C was significantly different from group A and group B. It indicated that the increase in A/O series is helpful to increase the microbial abundance in the reaction tanks, but it reduces the diversity difference among the reaction tanks in the process as well. This fact could be related to the impact load and residence time of the reactor. The relationship between impact load (indicated by P) and residence time (indicated by H) among the three groups is P_A_ > P_B_ > P_C_, H_A_ < H_B_ < H_C_, which is closely correlated to the relationship between the microbial diversity, i.e., a lower P and higher H corresponds to a larger S.

Figure 7c shows the Rank Abundance analysis of the sample, which can intuitively reflect the richness of the microbial species contained in the sample and the uniformity of the microbial community distribution. The abscissa span of samples A1 and B1 is found to be the smallest, and that of sample C5 is the largest, indicating that the species abundance of samples A1 and B1 is low, while the species abundance of sample C5 is the highest. This result is consistent with the results shown in the Simpson dilution curve of the alpha diversity index and the Heatmap chart of the microbial community structure (Figure 7d).

## 4. Conclusions

The effluent quality, membrane fouling and the microbial community structure of A/O-MBR, 2(A/O)-MBR, and 3(A/O)-MBR processes were investigated in this study. The results are as follows:(1)The effluent quality of the 2(A/O)-MBR process is better than that obtained with the other two processes. The average removal rates of COD, NH${}_{4}^{+}$-N, TP and TN are 95.29%, 89.47%, 83.55% and 78.58%, respectively.(2)The A/O-MBR process suffers from the highest membrane fouling, while the 3(A/O)-MBR process shows the lowest. Considering the effluent quality and cost issues, the 2(A/O)-MBR process can be selected in practical applications.(3)The analysis of the microbial community structure shows that the samples from this study have the largest abundance of γ-Proteobacteria, and Thaurea in the Proteobacteria is the key bacteria genus that dominates and affects the degradation of ammonia nitrogen and total nitrogen in the reactor.

In conclusion, this study demonstrates the feasibility of the multi-stage A/O-MBR system in enhancing wastewater treatment efficiency, and identifies the corresponding change in microbial community structure. The results, together with future works exploring in-depth analyses on reactor size and hydraulic retention time optimization, can provide guidance for efficient wastewater treatment equipment with smaller footprint.

## Figures and Tables

**Figure 1 membranes-12-00377-f001:**
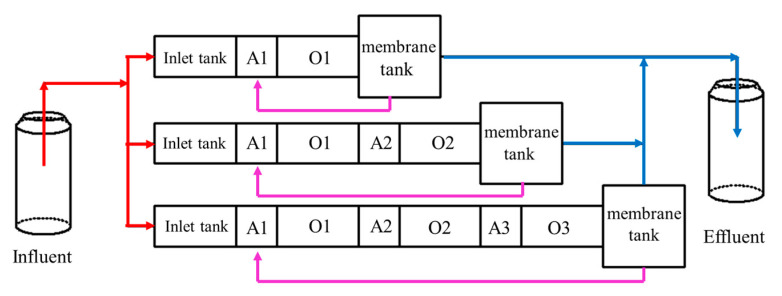
Overflow O-MAO-MBR process flow chart.

**Figure 2 membranes-12-00377-f002:**
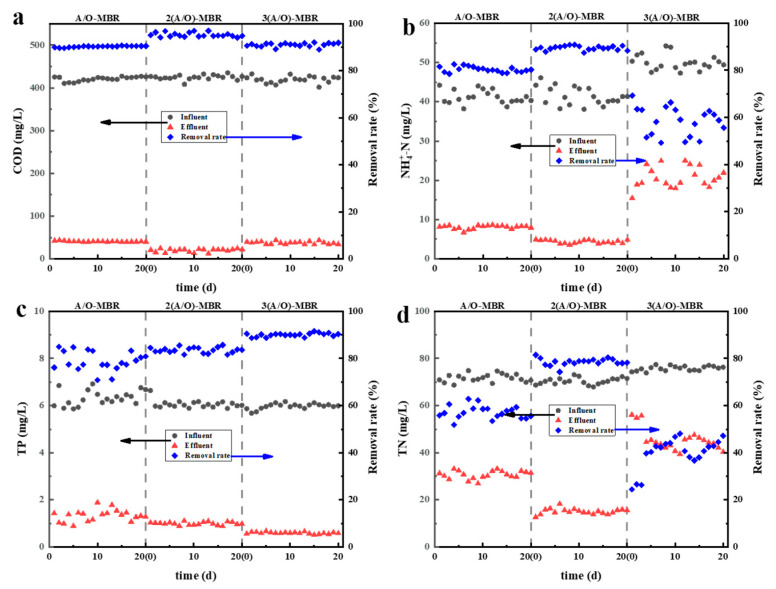
Treatment effect of overflow O-MAO-MBR on sewage (**a**) COD change, (**b**) NH${}_{4}^{+}$-N changes, (**c**) TP changes, and (**d**) TN changes.

**Figure 3 membranes-12-00377-f003:**
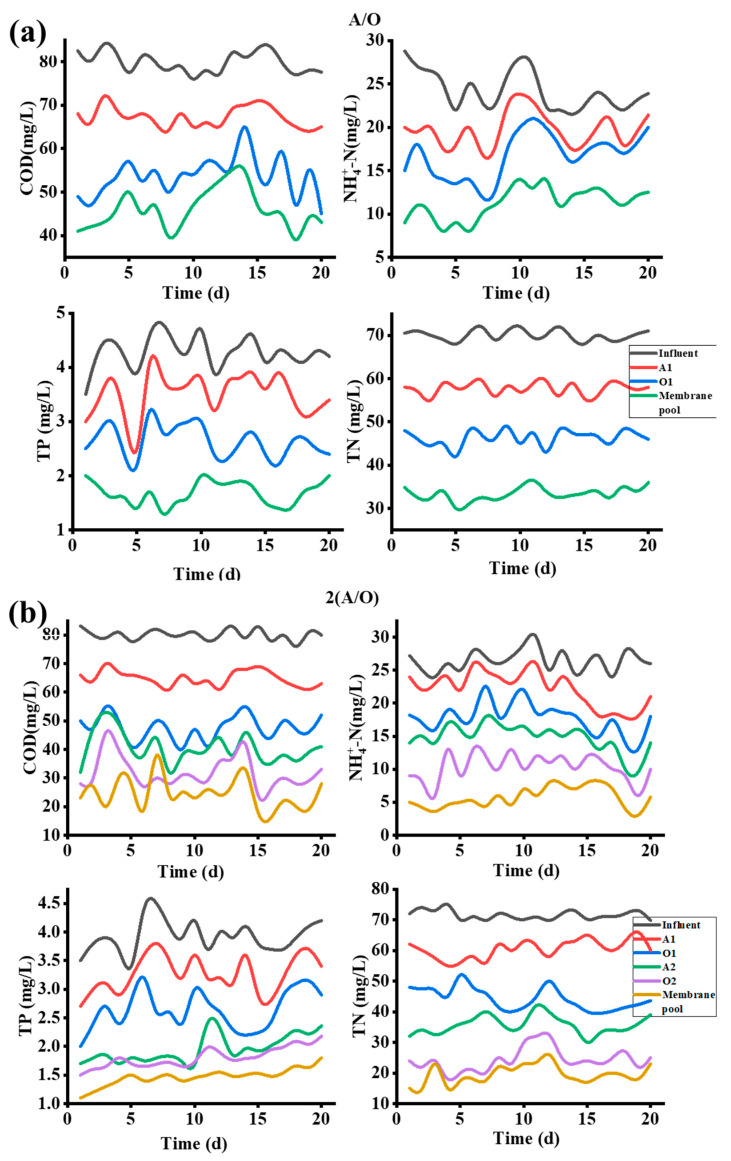
Treatment effect of overflow O-MAO-MBR on sewage (**a**) A/O-MBR change, (**b**) 2(A/O)-MBR changes, and (**c**) 3(A/O)-MBR changes.

**Figure 4 membranes-12-00377-f004:**
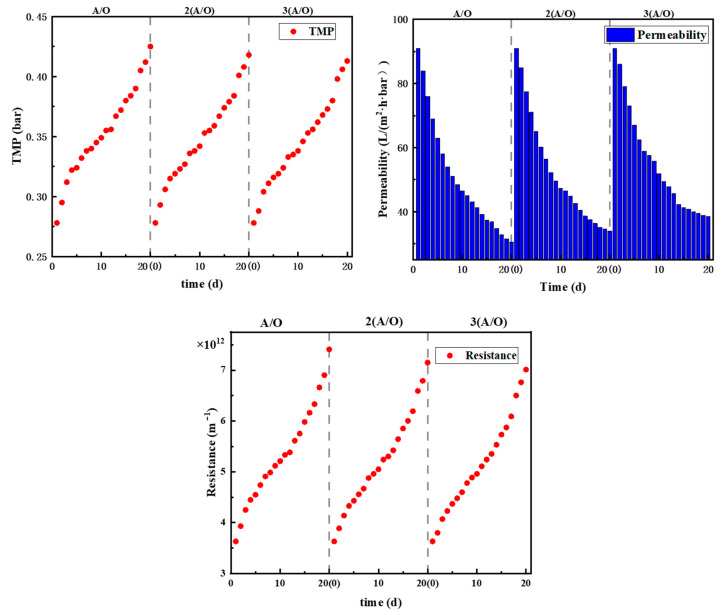
Changes of TMP, permeability and overall resistance with time in each process.

**Figure 5 membranes-12-00377-f005:**
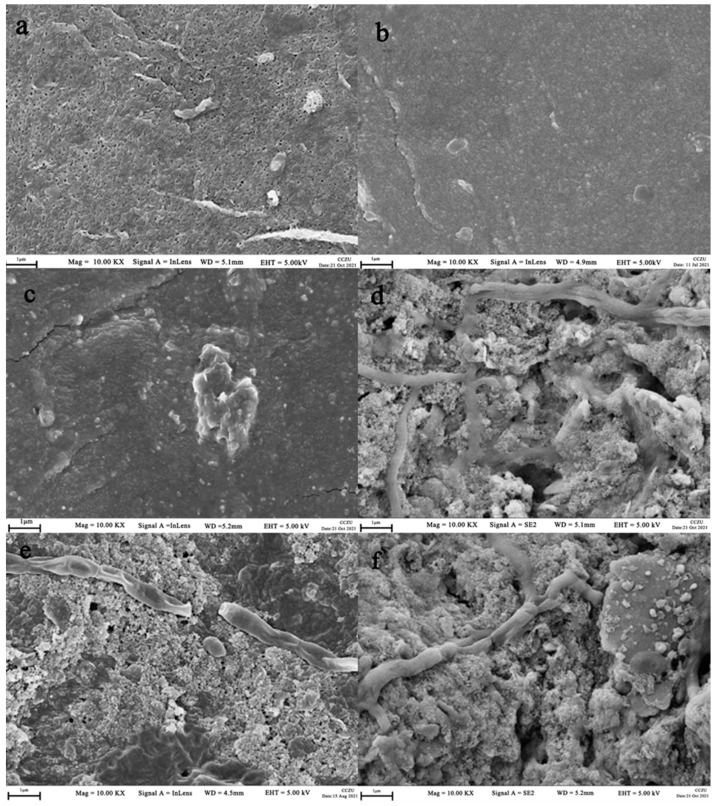
The SEM images of membrane: (**a**) the new membrane; (**d**) the newly contaminated membrane; (**b**) chemically cleaned membrane; (**e**) membrane fouled again after cleaning; (**c**) membrane cleaned for the second time; and (**f**) membrane fouled for the third time.

**Figure 6 membranes-12-00377-f006:**
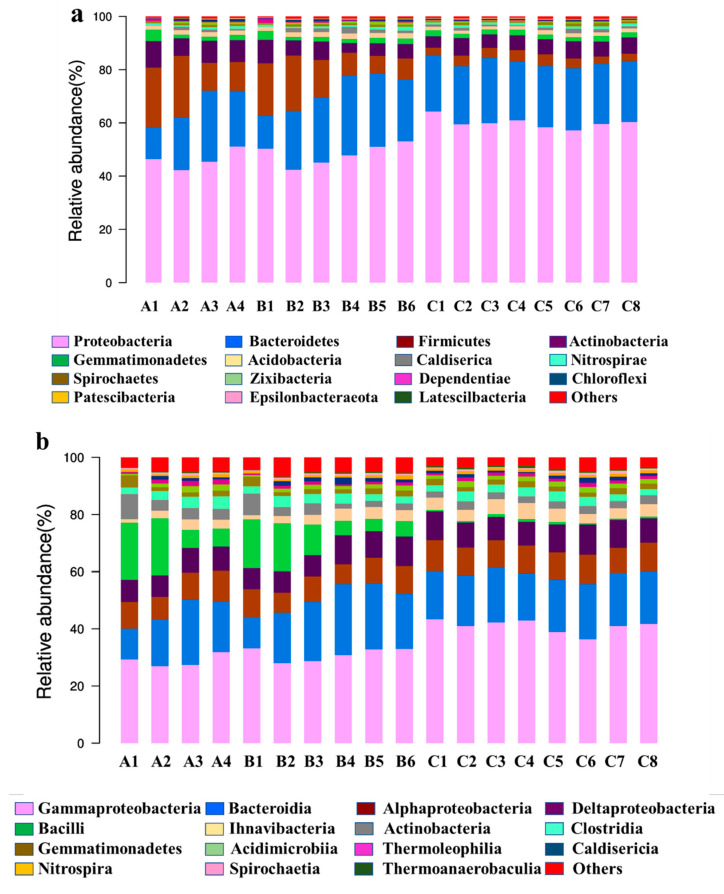
Microbial community structure histogram at the level of phylum, class, and genus for each sample: (**a**) phylum level; (**b**) class level; and (**c**) genus level.

**Figure 7 membranes-12-00377-f007:**
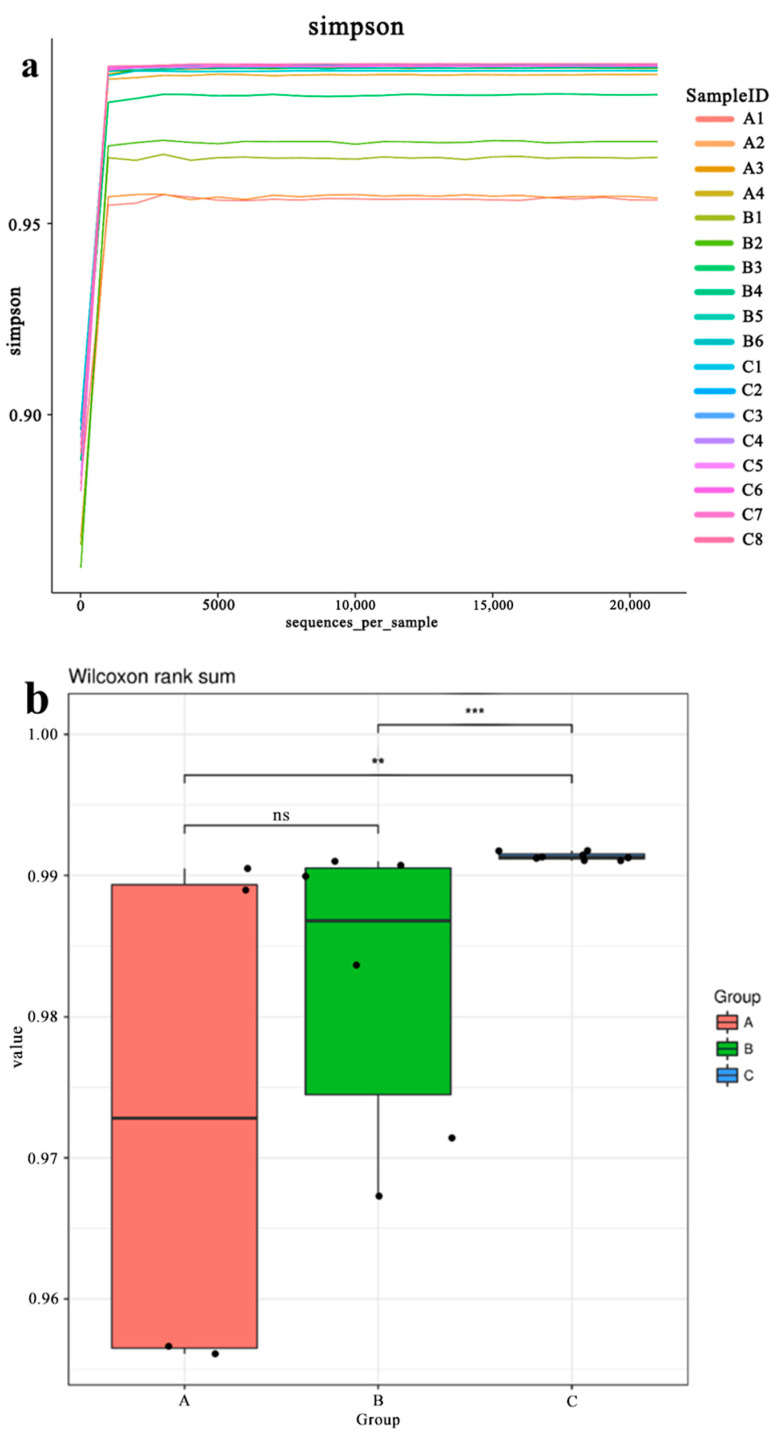
(**a**) Alpha diversity Simpson index dilution curve; (**b**) diversity index comparison boxplot chart; (**c**) Rank Abundance curve; (**d**) Heatmap chart of the microbial community structure.

**Table 1 membranes-12-00377-t001:** Main composition and concentration of simulated sewage.

Nutrient	Concentration (mg/mL)
C_6_H_12_O_6_	400.00
Urea	80.00
KH_2_PO_4_	25.00
NaCl	200.00
MgSO_4_·7H_2_O	4.15
FeSO_4_·7H_2_O	0.70
ZnSO_4_·7H_2_O	0.23
CoCl_2_·6H_2_O	0.34
MnSO_4_·H_2_O	0.18
Peptone	35.00
Beef extract	35.00
CaCl_2_	0.83

**Table 2 membranes-12-00377-t002:** Microbiological sample number.

Operation Process	The Reaction Cell to Which the Sample Belongs	Serial Number
A/O-MBR process	inlet pool, A1, O1, membrane pool	A1, A2, A3, A4
2(A/O)-MBR process	inlet pool, A1, O1, A2, O2, membrane pool	B1, B2, B3, B4, B5, B6
3(A/O)-MBR process	inlet pool, A1, O1, A2, O2, A3, O3, membrane pool	C1, C2, C3, C4, C5, C6, C7, C8

**Table 3 membranes-12-00377-t003:** Statistical data of the alpha diversity index.

Sample	Chao1	Goods_Coverage	Observed_Species	Shannon	Simpson
A1	1800.07	0.98	1282.40	7.14	0.96
A2	2588.13	0.97	1738.80	7.76	0.96
A3	2671.91	0.97	1851.70	8.43	0.99
A4	2458.15	0.97	1746.80	8.46	0.99
B1	1867.12	0.98	1330.40	7.38	0.97
B2	2636.38	0.97	1814.40	8.08	0.97
B3	2494.52	0.97	1803.20	8.27	0.98
B4	2625.58	0.97	1856.40	8.46	0.99
B5	2540.91	0.97	1787.20	8.41	0.99
B6	2354.33	0.97	1659.40	8.29	0.99
C1	2411.03	0.97	1678.10	8.29	0.99
C2	2478.51	0.97	1786.00	8.49	0.99
C3	2401.42	0.97	1668.80	8.32	0.99
C4	2407.11	0.97	1719.00	8.35	0.99
C5	2633.06	0.97	1915.70	8.57	0.99
C6	2527.03	0.97	1840.90	8.51	0.99
C7	2534.66	0.97	1801.70	8.44	0.99
C8	2500.06	0.97	1761.20	8.45	0.99

## Data Availability

The data presented in this study are available on request from the corresponding author.

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
