# Peer review of "Designing Multi-Stage 2 A/O-MBR Processes for a Higher Removal Rate of Pollution in Wastewater"

_membranes, 2022, doi:10.3390/membranes12040377_

Round 1
Reviewer 1 Report
Manuscript: The effects on wastewater treatment efficiency by multi-stage 2 A/O-MBR processes
Zhengzhong Zhou,* Bin Zhang, Qian Wang, Xiaoshan Meng, Qigang Wu, Tao Zheng, Taoli Huhe
This study evaluates feasibility of multistage A/O-MBR system in enhancing the wastewater treatment efficiency, and identifies the corresponding change in microbial community structure. The COD, TN and TP removal and membrane fouling study were performed.
The manuscript is well written and organised. Please see specific comments.
Line 51: “membrane pollution is serious.” Membrane fouling should be word that is more appropriate. It should be used throughout the MS.
Table 1. Main composition and concentration of simulated domestic sewage. Pl. provide nutrient name and chemical formula in-text (prior to Table). Pl. show COD, NH4, TN, and TP concentration of simulated wastewater. What is C:N:P ratio of simulated wastewater?
Line 89-90: The volume ratio of each tank is as follow, the inlet tank: anoxic tank : aerobic tank : membrane tank : spare tank = 2 : 1 : 2 : 4 : 1. What is logic for these ratios? What should be theoretical ratio of anoxic to aerobic tank to achieve complete Nitrogen removal?
Line 107-108: The sludge retention time (SRT) was 20 d. For aerobic or anaerobic or both? Since both tank has different MLSS? What was MLVSS/MLSS ratio?
Line 110-111: The DO of the anaerobic tank was kept below 0.3 mg/L, and the DO of the inlet tank was in the range of 1-2 mg/L. Have you used any anaerobic tank?
Line 183-184: Carbon source in the latter reaction tanks may be too low for the denitrification process, resulting in poorer total nitrogen removal. What should be ideal C:N ration for denitrification to occur? What was the C:N in each of the processes in your work (A/O, 29A/O) and 3 (A/O) respectively?
Line 186-187: ‘interrupting the anaerobic environment and impacting the denitrification process.’ Denitrification does not required anaerobic environment. It requires anoxic environment. DO concentration clearly shows it is anoxic condition? Pl. Correct.
Line 189-196: Phosphorous removal discussed. Biological P removal requires anaerobic tank. To remove both TN and TP anaerobic, anoxic and aerobic treatment train is essential. How did you achieve high P removal under anoxic conditions? Pl. explain in detail. Like DO, ORP is another important parameter to observe anoxic and anaerobic conditions in bioreactor which is usually measured.
Line 288-297: It has been mentioned that Thauera were important denitrifying bacteria. Is 1.54% Thauera is enough to achieve maximum TN removal as shown in this study (2(A/O) process?
Author Response
Dear reviewer,
Please see the attachment, thanks.

Reviewer 2 Report
Dear Editor,
Regarding this manuscrip, I judge it can be published after minor corrections (please see bellow). I have not checked plagiarism and self-citation issues.
Sincerely yours.
Minor suggestions:
The title does not portrait the findings of the work. How about “The effects on wastewater treatment efficiency by multi-stage 2 A/O-MBR processes designs”?
L14: please specify which standards, something as “with the Chinese sewage discharge standard”.
L25 and so on: Is “domestic” sewage a pleonasm?
L123: Please rephrase.
L137: An extra dot here “L m−2·h−1”
L186: Please standardize mg/L or mg L-1 in the whole text.
Figure 2: What is the purpose of narrows in this figure?
Figure 3: Captions are unclear inside this figure.
Figure 5: Sorry if I was wrong, why TOP in uppercase? Are 15 or 18?
Author Response
Dear reviewer,
Please see attachment, thanks.

Round 2
Reviewer 1 Report
After revision, I consider this MS is ok for publishing.
Author Response
Dear Reviewer,
Thank you for your comments.